# Ultrastructure of Ejaculatory Ducts of *Cerapanorpa nanwutaina* and *Furcatopanorpa longihypovalva* (Mecoptera: Panorpidae)

**DOI:** 10.3390/insects13111074

**Published:** 2022-11-21

**Authors:** Qi-Hui Lyu, Qing-Xiao Chen, Ya-Lan Sun, Wen-Jie Dong

**Affiliations:** 1College of Horticulture and Plant Protection, Henan University of Science and Technology, Luoyang 471023, China; 2Ultrastructural Pathology Diagnosis and Research Center, The First Affiliated Hospital of Henan University of Science and Technology, Luoyang 471003, China

**Keywords:** ejaculatory duct, male reproductive system, Panorpidae, morphology

## Abstract

**Simple Summary:**

Panorpidae are the most speciose family in Mecoptera and are commonly called scorpionflies, because the genital bulb of their males is enlarged and recurved upward, superficially resembling the stinger of a scorpion. The ejaculatory ducts of Panorpidae are studied for the first time via light and transmission electron microscopy. The ejaculatory duct consists of a median duct and an accessory sac. The epithelium of the median duct has conspicuous secretory activities, which are likely considered a supplement to the male accessory glands. The median duct of *F. longihypovalva* has one thin muscular layer, which is missing in that of *C. nanwutaina*. The thin or absent muscular layer may be associated with the sperm pump.

**Abstract:**

The ultrastructure of the ejaculatory duct was investigated in the scorpionflies *Cerapanorpa nanwutaina* (Chou 1981) and *Furcatopanorpa longihypovalva* (Hua & Cai, 2009) (Mecoptera: Panorpidae) using light and transmission electron microscopy. The ejaculatory ducts of both species comprise a median duct and an accessory sac. The median duct consists of a basal lamina, a mono-layered epithelium, a subcuticular cavity, and an inner cuticle. The accessory sac contains a single layer of epithelium and a basal lamina. A muscular layer is present in the accessory sac of *C. nanwutaina* and in the median duct of *F. longihypovalva*. The epithelia in the median duct and the accessory sac are well developed, their cells containing numerous cisterns of rough endoplasmic reticulum, mitochondria, and microvilli. The secretions of the median duct are first extruded into the subcuticular cavity and then into the lumen through an inner cuticle, while the secretions of the accessory sac are discharged directly into the lumen. The ejaculatory duct of *F. longihypovalva* is longer and has thicker epithelium with more cell organelles and secretions than that of *C. nanwutaina*.

## 1. Introduction

The ejaculatory duct is a median tube that transports the sperm from the vasa deferentia to the gonopore in most insects [1]. It is ectodermal in origin in the majority of insects and consists of a mono-layered epithelium and an inner cuticle [2]. However, it originates from both the ectoderm and mesoderm in all Lepidoptera, the parasitoid wasp *Diadromus pulchellus* (Hymenoptera: Ichneumonidae), the lovebug *Plecia nearctica* (Diptera: Bibionidae), and the southern green stink bug *Nezara viridula* (Hemiptera: Pentatomidae) [3].

The main function of the ejaculatory duct is to transport the sperm and the spermatophores to the gonopore [4]. More specifically, the ejaculatory duct has additional functions, such as the formation of spermatophore and seminal fluid in the midge *Chironomus plumosus* (Diptera: Chironomidae) [5] and in most Lepidoptera and Coleoptera [6], sperm activation in some lepidopterans [7,8,9], and the protection of the gametes and the reproductive tract of both sexes from bacterial infection in the fruit fly *Drosophila melanogaster* [10]. In addition, the ejaculatory duct also functions to enhance female fecundity and inhibits sexual receptivity in the housefly *Musca domestica* (Diptera: Muscidae) [11] and the stable fly *Stomoxys calcitrans* (Diptera: Muscidae) [12].

The ejaculatory ducts vary in number and morphology among different insect species. The duct is usually one and assumes a median tubular shape [1]. Ephemeroptera, however, have no ejaculatory duct so that a pair of vasa deferentia opens separately to the exterior [13,14,15]. In Dermaptera [16], on the other hand, there are two ejaculatory ducts, one of which is degenerated in Spongiphoridae, Chelisochidae, and Forficulidae [17]. More complicatedly, the duct consists of a wider upper and a cylindrical lower duct connected with a funnel-like constriction in the African migratory locust *Locusta migratoria migratorioides* (Orthoptera, Acrididae) [18]. The ejaculatory duct is crozier-shaped with the hook end enlarged anteriorly and subspherical swelling posterior in the milkweed bug *Oncopeltus fasciatus* (Heteroptera: Lygaeidae) [19]. The anterior part of the duct is subdivided into seven sections, each with its own ultrastructural characteristics in the skipper butterfly *Calpodes ethlius* (Lepidoptera: Hesperiidae) [20]. The ejaculatory duct comprises a muscular sheath, a monolayered epithelium, and an inner cuticle in many insects [1]. The morphology of the ejaculatory duct, however, has only been briefly described in Mecoptera [21].

Panorpidae, the most speciose family within Mecoptera [22], have diverse mating patterns [23,24,25]. Before copulation, the male of Panorpidae may offer a prey or a salivary mass as a nuptial gift [23,24]. Alternatively, they may adopt a coercive mating tactic using their complex grasping structures [26,27,28], such as the notal organ, postnotal organ, anal horn(s), epandria, hypandria, gonostyli, and parameres. During mating, male scorpionflies directly transfer liquid sperm to the female through the advance and return movement of the sperm pump [21]. Previous studies on the ejaculatory duct of Panorpidae have only been conducted on morphology and histology in *Panorpa liui* [21], but study of the ultrastructural morphology is still lacking. Therefore, we investigated the ultrastructure of the ejaculatory ducts of the scorpionflies *Cerapanorpa nanwutaina* (Chou 1981) and *Furcatopanorpa longihypovalva* (Hua & Cai, 2009) using light and transmission electron microscopy.

## 2. Materials and Methods

### 2.1. Collection of Specimens

Male adults of *C. nanwutaina* were collected from the Huoditang Forest Farm in the Qinling Mountains, Shaanxi Province, central China, in late July 2017. Male adults of *F. longihypovalva* were captured from Huanglongtan in the Hualong Mountains, Shaanxi Province, central China, in early July 2017. These species were determined from Chou et al. (1981) [29] and Hua and Cai (2009) [30].

### 2.2. Gross Anatomy

Live adults were anesthetized with diethyl ether [31]. The ejaculatory duct was dissected out rapidly. Photographs were taken with a QImaging Retiga2000R Fast 1394 Digital CCD equipped on a Nikon SMZ168 stereomicroscope (Nikon, Tokyo, Japan).

### 2.3. Transmission Electron Microscopy (TEM)

The ejaculatory ducts were fixed with a mixture of 2.0% paraformaldehyde and 2.5% glutaraldehyde in phosphate-buffered saline (PBS, 0.1 M, pH 7.2) at 4 °C for 8 h. Then, the samples were rinsed with PBS six times, and post-fixed with 1% osmium tetroxide (OsO_4_) at 4 °C for 2 h. After rinsing six times with PBS, the samples were dehydrated through a graded series of acetone. The samples were infiltrated successively through a graded mixture of acetone and Epon 812 resin before being transferred to pure Epon 812 resin. The samples were finally polymerized in pure Epon 812 resin at 30 °C for 24 h and 60 °C for 48 h.

The resin-embedded ejaculatory ducts were cut into ultrathin sections (60 nm) using a diamond knife on a Leica EM UC7 ultramicrotome (Leica, Nussloch, Germany). Ultrathin sections were double stained and examined in a Tecnai G2 Spirit Bio Twin transmission electron microscope (FEI, Hillsboro, OR, USA).

### 2.4. Light Microscopy (LM)

For histological observations, the embedded samples were cut into semi-thin sections (1 μm) with a glass knife on the same ultramicrotome, stained with 1% toluidine blue, and examined under a Nikon Eclipse 80i light microscope (Nikon, Tokyo, Japan) mounted with a digital camera Nikon DS-Ri1 (Nikon, Tokyo, Japan).

## 3. Results

The male reproductive systems of *C. nanwutaina* and *F. longihypovalva* are similar in general structures, comprising a pair of testes, a pair of vasa deferentia, and an ejaculatory duct [31,32]. Each vas deferens is coiled for its distal part to form an epididymis and enlarged in its medio-posterior portions to form a seminal vesicle. The ejaculatory duct connects the posterior end of post-vesicular vasa deferentia and the base of the sperm pump in the nineth abdominal segment.

### 3.1. Gross Morphology of the Ejaculatory duct

The morphology of the ejaculatory duct is evidently diverse between *C. nanwutaina* and *F. longihypovalva* (Figure 1 and Figure 2).

The ejaculatory duct consists of a median duct and a symmetrical accessory sac in *C. nanwutaina* (Figure 1A,B). The median duct is milky white and approximately 0.23 mm in overall diameter and 0.61 mm in length (Figure 1A,B). The diameter of the duct increases gradually from the basal region to the middle region and diminishes gradually toward the distal end (Figure 1A,B). The symmetrical accessory sac is formed by the wall of the median duct extending backward and upward from lateral side. The sac occurs at the distal one-third of the duct and extends caudally beyond the apex of duct (Figure 1B).

The ejaculatory duct comprises a median duct and an accessory sac, which encloses the median duct in *F. longihypovalva*. The median duct is approximately 1.18 mm long and is subdivided into two portions (Figure 2A). The basal one-sixth is much thinner, half as wide as the distal part. The diameter of the distal part increases sharply at the junction with the basal portion and decreases slightly toward the distal end (Figure 2B,C). The accessory sac is translucent, and its smooth inner surface tightly covers the distal part of the median duct (Figure 2A). The outer surface of the accessory sac exhibits numerous sphere-shaped structures, which are formed by the bending of the wall (Figure 2D).

### 3.2. Histology and Ultrastructure of the Ejaculatory Duct in C. nanwutaina

The median duct of the ejaculatory duct has no significant differences ultrastructurally at various regions. It consists of a basal lamina, a mono-layered epithelium, a subcuticular cavity, and an inner cuticle (Figure 3). In the basal region of epithelial cells, the plasma membranes have conspicuous invaginations, forming distinct intercellular spaces (Figure 3B,C). The secretory epithelial cells exhibit straight courses and occupy nearly half of the volume of the median duct in the intermediate region with abundant microvilli (Figure 3A,B). The irregular nuclei are situated in the basal region of the cells (Figure 3C). Numerous cisterns of rough endoplasmic reticulum and mitochondria are distributed in the whole cytoplasm (Figure 3D,E). Some electron-dense secretory granules are visible in the intermediate and apical regions (Figure 3D,H). The apical plasma membranes have an irregular shape (Figure 3G). Packed aligned microvilli are found on the apical cell surface and fill the deep sunk of the epithelium (Figure 3A,F). The subcuticular cavity is wide, occupying most of the remaining volume, with abundant electron-dense granules and some electron-dense vesicles (Figure 3F,H). The narrow lumen is enclosed by a thick inner cuticle.

The symmetrical accessory sac consists of a muscular sheath and a monolayered epithelium surrounded by a basal lamina (Figure 4A,B). Two layers of epithelia closely adhere to the median duct, with poorly developed several layers of muscle cells between them and 7–9 well-developed layers of muscle cells outside (Figure 4A–D). The epithelium adjacent to the median duct is closely pressed, and the epithelium adjacent to the hemocoel is interdigitated in the basal region, with distinct intercellular spaces (Figure 4B). The flat nuclei occupy most of cytoplasm in the basal region (Figure 4D,E). The cytoplasm is rich in mitochondria and rough endoplasmic reticulum (Figure 4F,G). The microvilli of various lengths extend into the central lumen with several granules dispersed among them (Figure 4G,H). Several layers of muscle cells are visible between the symmetrical accessory sac and the median duct, with abundant tracheoles and hardly any epithelium (Figure 4A,B,D).

### 3.3. Histology and Ultrastructure of the Ejaculatory duct in F. longihypovalva

The median duct of *F. longihypovalva* has one muscular layer, which is lacking in that of *C. nanwutaina* (Figure 3A and Figure 5A–C). The basal plasma membranes have numerous infoldings and generate a labyrinth. The cytoplasm of this part is considerably scarce (Figure 5D). The cytoplasm is mainly concentrated in the middle part and contains oval nuclei, abundant mitochondria, and some rough endoplasmic reticulum (Figure 5A). The adjacent cells are held together by zonula adherens and septate junctions in the apical region (Figure 5E). Extensive packed microvilli occur on the apical cell surface, with some microtubules scattered around the basal part (Figure 5B,E,F). Short microvilli bundles fill the apical intracellular space and project into the subcuticular cavity, while longer microvilli bundles extend into the central epithelial layer, as deep as near the nuclei (Figure 5B). Two types of secretions are found; abundant electron-dense granules are mainly located between the basal microvilli, and electron-lucent vesicles are scattered in the apical and basal regions (Figure 5E,G). The subcuticular cavity contains abundant secretions. The lumen is flat and narrow, surrounded by a zigzag inner cuticle in cross section (Figure 5H).

The accessory sac of *F. longihypovalva* consists of a basal lamina and an epithelium (Figure 6A). The plasma membranes conspicuously expand out toward the periphery in the basal one-fifth of the epithelium adjacent to the hemocoel (Figure 6B). The nuclei are irregular and situated in the basal or medial region of the cells (Figure 6B,C). The extensive microvilli project into the medial epithelium and are clustered at the apical surface of the epithelium (Figure 6B,D), with abundant mitochondria (Figure 6E), vesicles, secretory lumps, and rough endoplasmic reticulum (Figure 6F). The flat epithelium is adjacent to and tightly presses the median duct (Figure 6G,H). A thin muscular layer is present between the accessory sac and the median duct. Only a few tracheoles are visible in the muscular layer (Figure 6H). 

## 4. Discussion

In Mecoptera, the gross morphology of ejaculatory ducts have been investigated in four families (Boreidae, Meropeidae, Bittacidae, and Panorpidae) [33,34,35,36,37]. The ejaculatory ducts of Panorpidae are short and straight, and have a conspicuous accessory sac or two [34]. The ejaculatory duct of *F. longihypovalva* is relatively longer and has a thicker epithelium, whereas the epithelium is thinner and has a large lumen in *C. nanwutaina*. In addition, the ejaculatory duct of *F. longihypovalva* has more mitochondria, rough endoplasmic reticulum, and secretions than that of *C. nanwutaina*. These ultrastructural features suggest that the epithelium of *F. longihypovalva* might afford more secretory functions [2]. This inconsistency may be caused by the differences in the number and size of the ectodermal accessory glands between these two species.

The male of *C. nanwutaina* possesses five pairs of ectodermal accessory glands [31], while the male of *F. longihypovalva* only has one pair [32]. The accessory glands have a variety of functions to help males successfully mate in most insects [11,38]. When lacking accessory glands, the males usually produce seminal fluid by the ejaculatory duct as in the midge *Chironomus plumosus* (Diptera: Chironomidae) [5] and in the housefly *Musca domestica* (Diptera: Muscidae) [11]. We argue that the ejaculatory ducts may have more active secretory activity as a supplement to the male accessory glands in *F. longihypovalva* than in *C. nanwutaina*.

The ejaculatory ducts have conspicuous secretory activities in panorpids, although there are some differences between the two species. Secretions are first released and accumulated in the subcuticular cavity and are then discharged into the central lumen through the inner cuticle without marked pores. Similar types of cells have been described as Class I glandular cells, simply contacting with the glandular cuticle [39,40]. The cells of the ejaculatory ducts are rich in rough endoplasmic reticulum, indicating that proteinaceous secretions might be produced [41,42]. The scarcity of smooth endoplasmic reticulum may imply that the nonproteinaceous material is scarcely synthesized, compared with the osmeterium glands or the adhesive glands [43,44]. In the previous studies, the usage of the terms “ejaculatory duct” and “ejaculatory sac” are quite inconsistent in Panorpidae. The common ectodermal exit tube between the 9th and 10th segments was referred as ejaculatory duct or ductus ejaculatorius by Grell (1942) [36] and Sinclair et al. (2007) [45], but was called an ejaculatory sac by Potter (1938) [34] and Shen and Hua (2013) [21]. Potter (1938) claimed that the ejaculatory duct refers to the symmetrical lateral sac, and the so-called ejaculatory sac indicates the median duct. Based on our investigations, there is no other organ between the dilated median duct and the sperm pump. The symmetrical lateral sac is located on both sides of the median duct and not directly connected to the sperm pump. Therefore, the so-called ejaculatory sac may be more appropriately called the ejaculatory duct. The common exit tube is called the ejaculatory sac because of its contraction function during ejaculation [21]. With the same function, the common exit tube is still called the ejaculatory duct by Hünefeld and Beutel (2005) [46] and Sinclair et al. (2007) [45]. Thus, we regard the common exit tube as the ejaculatory duct.

The muscular layer is absent in the median duct, but is present in the accessory sac of the ejaculatory duct of *C. nanwutaina*, while the opposite is observed in *F. longihypovalva*. Significantly, the median duct only has one thin muscular layer in *F. longihypovalva*. Usually, at least a part of the wall of the ejaculatory ducts is muscular in most insects [47,48,49]. However, the ejaculatory duct has no muscular layers in *Apis* and some *Melipona* [50,51]. The absence of muscles is likely related to the sperm pump, through which the males directly transfer seminal fluid to the females in most males of Mecoptera, Siphonaptera, and Diptera [46]. During copulation, the males mentioned above transfer the sperm by waving the piston of the sperm pump instead of squeezing the tube of spermatophore through the ejaculatory duct as in other species [21,52]. Thus, the thin or absent muscle layer of the ejaculatory duct may be associated with the sperm pump or propulsion apparatus in Panorpidae.

The male of *Cerapanorpa* usually provides a salivary secretion and copulate with the female in a V-shaped mating position. The male uses its notal organ, single anal horn, and genitalia to grasp the female wings, abdominal segment VIII (A8), and genital segments, respectively [27]. The male of *Furcatopanorpa* provides liquid salivary secretion and maintains a O-shaped mating position by continuously providing secretion in a mouth-to-mouth mode rather than clamping the female with non-genital grasping devices [53]. The nuptial feeding mating tactic is more effective than the coercive in copulation of Panorpidae [54,55]. The male of *Furcatopanorpa* likely obtains longer copulation duration compared with the male of *Cerapanorpa*. Since the lumen of the ejaculatory duct is small in *Furcatopanorpa*, the sperm may not be mainly stored in the ejaculatory duct, but rather in the epididymis and vasa deferentia far from the genitalia. The sperm storage and transfer of scorpionflies need further research.

## Figures and Tables

**Figure 1 insects-13-01074-f001:**
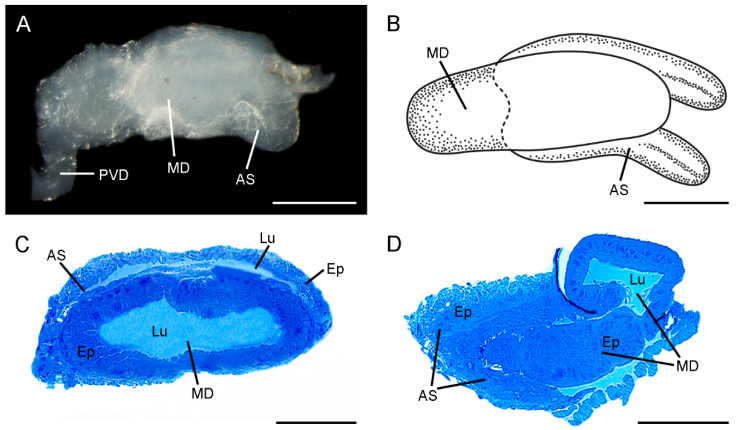
General morphology of the ejaculatory duct of *Cerapanorpa nanwutaina*. (**A**) Lateral view under LM. (**B**) Schematic diagram of habitus, ventral view. (**C**) Cross section showing the epithelia and lumens in the median duct and symmetrical accessory sac. (**D**) Longitudinal section of posterior part, showing that the symmetrical accessory sac partially encloses the median duct. AS, accessory sac; Ep, epithelium; MD, median duct; PVD, post-vesicular vas deferens. Scale bars: (**A**,**B**) = 200 μm; (**C**,**D**) = 100 μm.

**Figure 2 insects-13-01074-f002:**
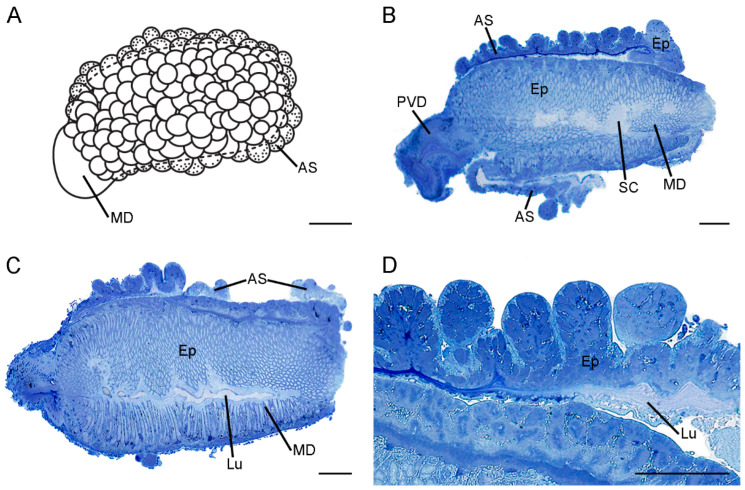
General morphology of the ejaculatory duct of *Furcatopanorpa longihypovalva*. (**A**) Schematic diagram of the habitus, showing the median duct and the accessory sac. (**B**) The accessory sac extends from ventral side of the median duct in the longitudinal section. (**C**) Longitudinal section showing the small lumen of the median duct. (**D**) A magnification of the chambers of accessory sac. AS, accessory sac; Ep, epithelium; Lu, lumen; MD, median duct; PVD, post-vesicular vas deferens; SC, subcuticular cavity. Scale bars: (**A**) = 200 μm; (**B**–**D**) = 100 μm.

**Figure 3 insects-13-01074-f003:**
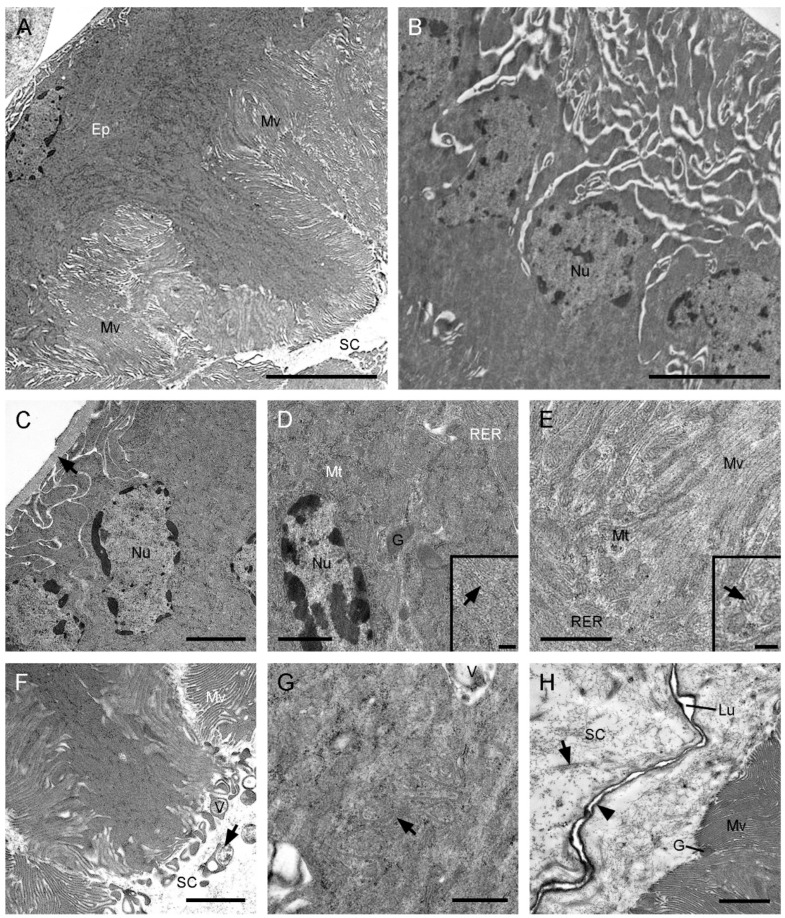
Median duct of ejaculatory duct of *Cerapanorpa nanwutaina*, cross-sections, TEM. (**A**) Epithelium and subcuticular cavity. (**B**) Basal region of the epithelium. (**C**) Nuclei in the basal region of the epithelium. Arrow points to the basal lamina. (**D**) Organelles and granules in medial cytoplasm. Inset shows the magnification of rough endoplasmic reticulum (arrow). (**E**) Basal part of microvilli with mitochondria and rough endoplasmic reticulum. Inset shows the magnification of mitochondrion (arrow). (**F**) Apical region of the epithelium, showing the microvilli and secretory fiber-bounded vesicle. Arrow points to the vesicle budding off into the subcuticular cavity. (**G**) Infolding (arrow) of the plasma membrane. (**H**) Subcuticular cavity and lumen. Arrow shows the electron-dense granule. Arrowhead points to the inner cuticle. Ep, epithelium; G, granule; Lu, lumen; Mt, mitochondrion; Mv, microvillus; Nu, nucleus; RER, rough endoplasmic reticulum; SC, subcuticular cavity; V, vesicle. Scale bars: (**A**,**B**) = 5 μm; (**C**,**F**,**H**) = 2 μm; (**D**,**E**,**G**) = 1 μm; insets = 0.25 μm.

**Figure 4 insects-13-01074-f004:**
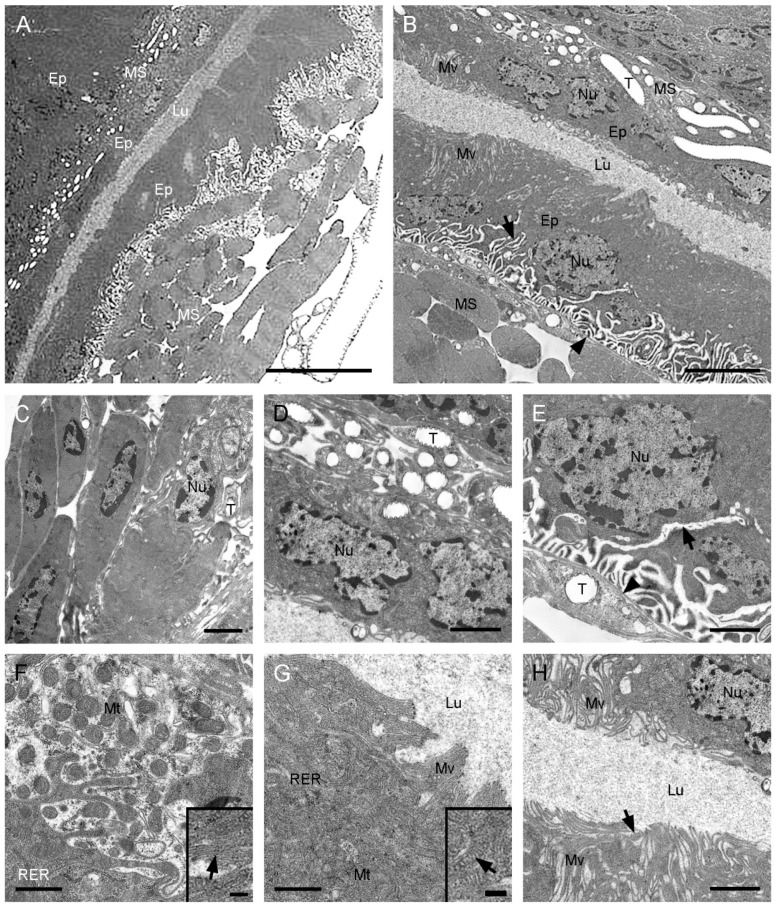
Accessory sac of the ejaculatory duct of *Cerapanorpa nanwutaina*, cross-sections, TEM. (**A**) Muscular layers surround the epithelium. (**B**) Two epithelia surround the lumen. Arrow points to the basal plasma membrane. Arrowhead points to the basal lamina. (**C**) Nuclei of muscle cell. (**D**) Nuclei of the inner side of epithelium. (**E**) Nuclei of the outside of epithelium. (**F**) Organelles in medial cytoplasm. Inset shows the magnification of rough endoplasmic reticulum (arrow). (**G**) Apical region of the epithelium. Inset shows the magnification of mitochondrion (arrow). (**H**) Secretions in lumen. Arrow shows the granule in intracellular space. Ep, epithelium; Lu, lumen; MS, muscular sheath; Mt, mitochondrion; Mv, microvillus; Nu, nucleus; RER, rough endoplasmic reticulum; T, tracheole. Scale bars: (**A**) = 10 μm; (**B**) = 5 μm; (**C**–**E**,**H**) = 2 μm; (**F**,**G**) = 1 μm; insets = 0.25 μm.

**Figure 5 insects-13-01074-f005:**
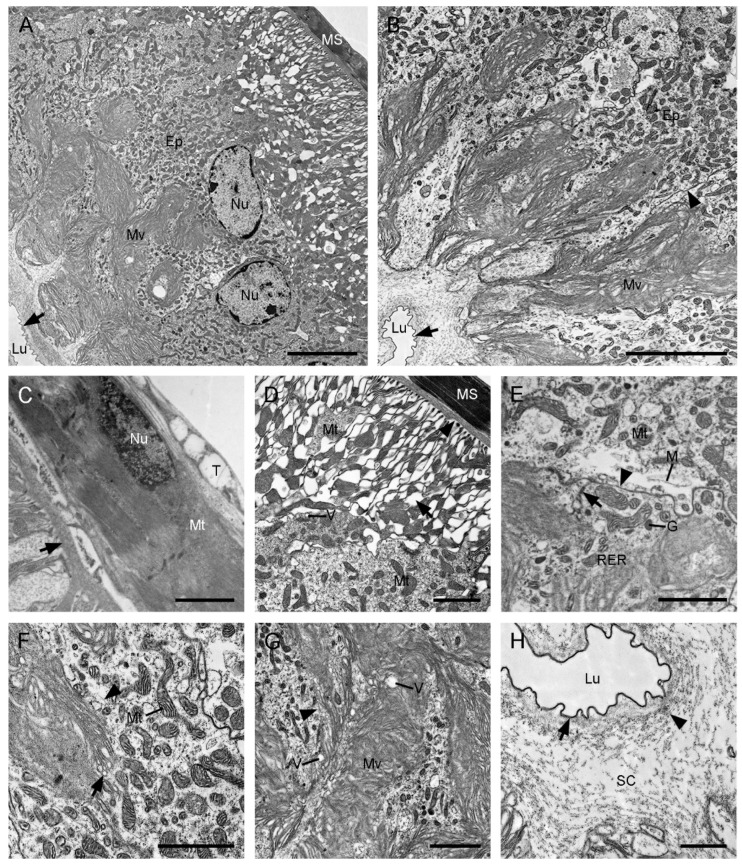
Median duct of the ejaculatory duct of *Furcatopanorpa longihypovalva*, cross-sections, TEM. (**A**) Muscular layers and epithelium. Arrow points to the inner cuticle. (**B**) Microvilli extend from the middle of the epithelium to near the central lumen. Arrowhead points to the plasma membrane. Arrow points to the inner cuticle. (**C**) Organelles and nucleus in muscle sheath. Arrow points to the basal lamina. (**D**) Basal part of the epithelium, showing mitochondrion. Arrow shows an intracellular space. Arrowhead points to the basal lamina. (**E**) Zonula adherens (arrow) and septate junction (arrowhead) in the apical region. (**F**) Organelles near by the base of the microvilli. Arrow points to the small vesicle. Arrowhead shows the microtubule. (**G**) Granules and vesicles around the microvilli. Arrowhead shows the microtubule. (**H**) Secretions in subcuticular cavity. Arrow shows the electron-dense granule. Arrowhead points to the inner cuticle. Ep, epithelium; Lu, lumen; M, microtubule; MS, muscular sheath; Mt, mitochondrion; Mv, microvillus; Nu, nucleus; RER, rough endoplasmic reticulum; SC, subcuticular cavity; T, tracheole; V, vesicle. Scale bars: (**A**,**B**) = 5 μm; (**C**–**G**) = 2 μm; (**H**) = 1 μm.

**Figure 6 insects-13-01074-f006:**
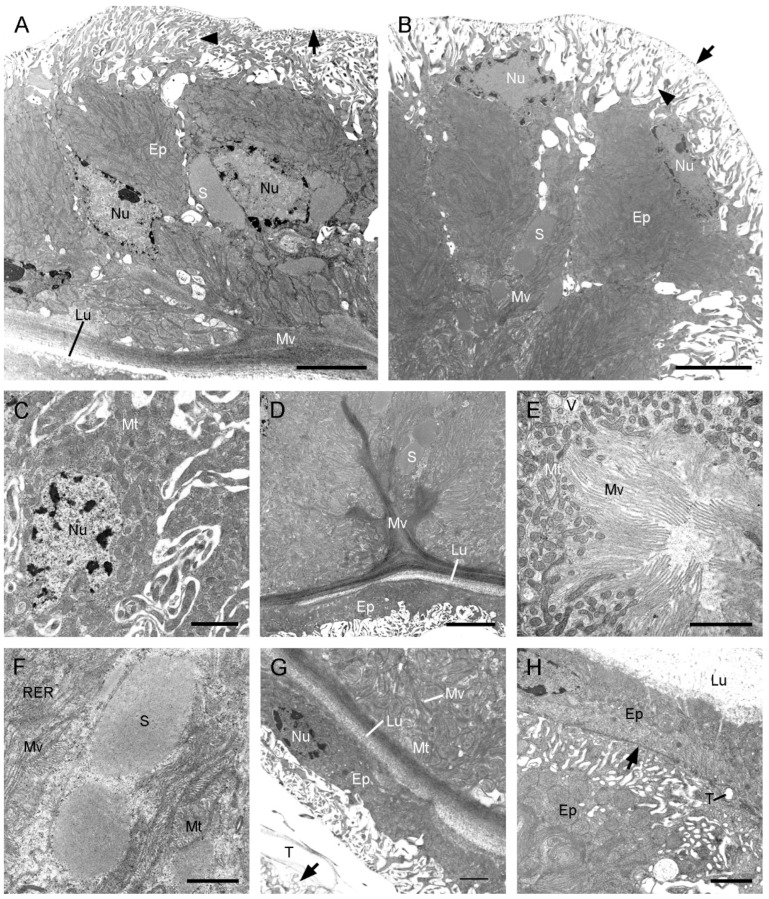
Accessory sac of the ejaculatory duct of *Furcatopanorpa longihypovalva*, cross-sections, TEM. (**A**) Epithelium. Arrow points to the basal lamina. Arrowhead points to the intracellular spaces. (**B**) Microvilli project into the medial epithelium with granules. Arrow points to the basal lamina. Arrowhead points to the intracellular spaces; (**C**) irregular nucleus and surrounding mitochondria. (**D**) Microvilli cluster. (**E**) Mitochondria near by the base of microvilli; (**F**) granules surrounded by microvilli and rough endoplasmic reticulum. (**G**) Inner side of the epithelium. Arrow shows the epithelium of the median duct. (**H**) Junction of accessory sac and the median duct. Arrowhead points to the muscular layer. Ep, epithelium; G, granule; Lu, lumen; Mt, mitochondrion; Mv, microvillus; Nu, nucleus; RER, rough endoplasmic reticulum; T, tracheole. Scale bars: (**A**,**B**,**D**) = 5 μm; (**C**,**E**,**G**,**H**) = 2 μm; (**F**) = 1 μm.

## Data Availability

The microscopy data supporting the conclusions of this manuscript will be made available by the authors, without undue reservation, to any qualified researcher.

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
