# Peer review of "Ultrastructure of Ejaculatory Ducts of Cerapanorpa nanwutaina and Furcatopanorpa longihypovalva (Mecoptera: Panorpidae)"

_insects, 2022, doi:10.3390/insects13111074_

Round 1

Reviewer 1 Report

The manuscript (MS) by Lyu et al., about the “Ultrastructure of ejaculatory ducts of Cerapanorpa nanwutaina and Furcatopanorpa longihypovalva (Mecoptera: Panorpidae)” is interesting, the results are appropriately, and images are of good quality, especially those of TEM. However, most of them are at low magnification, so it is impossible to recognize many of the cellular structures indicated in them. Furthermore, the images captured with the stereomicroscope (Figs. 1A, B and 2A) do not help the reader understand how the median duct and the accessory sac associate to form the ejaculatory duct. Therefore, I suggest the authors make a diagram of the ejaculatory duct showing the arrangement of the two structures that compose it. The scheme will undoubtedly make it easier for the reader to understand and compare this organ of these two species with other species. Anyway, I recommend that the MS be accepted for publication after the authors observe the above recommendations and the ones made in the attached pdf.

Reviewer 2 Report

The authors present a thorough investigation of the male reproductive systems of two species of scorpionflies, precisely on their ejaculatory ducts. The microscopy is straightforward and well presented. The outcomes are clearly presented. The results are well discussed in light of the literature and the own findings, however, I have troubles to follow how the authors explain the morphological differences between the two species based on their findings and in light of apparent behavioural differences between the sperm transfer of the two species. This should be sharpened in a minor revision. 

In general this study 

Main comments:

The introduction should also provide information on the mating behaviour and sperm transfer of Panorpidae. In the present state the literature on ejaculatory ducts in other insects is well presented, but the insects of concern are only briefly mentioned. What do we know abou the reproduction of Panorpidae and how does the presented study further the knowledge regarding their reproductive biology?

Discussion

The manuscript lacks a concluding discussion about the morphological differences between the species in regard to their mating behaviour. Why do these species differ in morphology and how can this be explained by their sperm transfer? This is discussed in comparison to other insect groups, but not within Panorpidae.  

Figure legends: 

Never use leading zeros. 0.2mm should be 200 µm

Few minor comments: 

l. 16 "is lack" should be "is missing"

l. 179 "side" should be plural ("sides")

l. 201 "is lack" should be "is lacking"

Reviewer 3 Report

The manuscript number 1973887, entitled “Ultrastructure of ejaculatory ducts of Cerapanorpa nanwutaina and Furcatopanorpa longihypovalva (Mecoptera: Panorpidae)” by Lyu and co-authors could contribute to the knowledge of the fine structure and function of an important tract of the male reproductive apparatus, the ejacutor duct, in two species of the family Panorpidae, not described so far, in which the epithelium has a secretory activity. The Introduction and M&M sections of the MS are well written (beside some wrong plurals such as “epitheliums” (abstract, line 7; caption of the Figures 1 and 4), or trivial adfirmations such as “non-cellular basal lamina”, Paragraph 3.2, line 4). In the discussion, data are well argumented and properly accompained by literature citations. Nevertheless, the Results section presents EM micrographs very poor in resolution, particularly Figures 3, 4 an 6. Due, most probably, to the fixation procedure not properly carried out, the cell ultrastructure details cited in the text and in the captions of the figures are very difficult to be  clearly observed.

Since this paper deals with the ultrastructural data, relevant to compare the secretory activity of the ejaculatory duct in two species of scorpionflies, the pictures should not be of low quality. For this reason, I can not suggest this paper to be published in “Insects”.

Round 2

Reviewer 3 Report

The revised version of te MS number 1973887 has been added with some interesting information on the reproductive behavior in Panorpidae, as suggested by one reviewer. However, the last part of the discussion on sperm transfer in the two species (lines 306-309), based on Author’s assumtions on an inverse relationship between the duration of copulation and the size of the lumen of the ejaculatory duct appear to be too “affirmative” and not directly supported by the results of the present paper. Authors should better explain and change the text accordingly.

Although some small changes have been done on the images (i.e. insets in Figures 3 and 4; lettering colour in Figure 4), their quality has not been enhanced.

On the other hand, since I suspect some problem in the fixation procedure, the entire TEM preparation shoud have been performed again.  However, I understand that insect specimens could be not avilable anymore to repeat the work.

Overall, the paper can be published, after the changes suggested in the discussion section and addressing the following minor comments.

 Minor comments:

line 17 “Their thin or absent of muscular...“ should be “Their thin or absent muscular...“

line 33 “..an median tube..” should be “…a median tube”.

Line 40 “…spermand…” should be “ …sperm and…”

Line 64 “Panorpidae is the most speciose family within Mecoptera [22], have diverse mating patterns [23–25].” should be

“Panorpidae, the most speciose family within Mecoptera [22], have diverse mating patterns [23–25].”

Line 164 “(E) Basal part of microvilli with mitochondrion and rough endoplasmic reticulum.” Should be “(E) Basal part of microvilli with mitochondria and rough endoplasmic reticulum.”

Line 200 …”zonula adheren” should be “zonula adherens”
